# Precision Isolation of Circulating Leukemia Cells in Chronic Myelogenous Leukemia Patients Using a Novel Microfluidic Device and Its Clinical Applications

**DOI:** 10.3390/cancers15235696

**Published:** 2023-12-03

**Authors:** Dongfang Ouyang, Ningxin Ye, Kun Yang, Yiyang Wang, Lina Hu, Shuen Chao, Mehmet Toner, Yonghua Li

**Affiliations:** 1Center for Engineering in Medicine and Surgery, Massachusetts General Hospital, Harvard Medical School, Charlestown, Boston, MA 02129, USA; 2Shriners Hospital for Children, Boston, MA 02114, USA; 3Department of Biology, University of Ottawa, Ottawa, ON K1N 6N5, Canada; 4Department of Mechanical & Industrial Engineering, University of Toronto, Toronto, ON M5S 3E8, Canada; 5Department of Microbiology, Immunology and Molecular Genetics, University of California Los Angeles (UCLA), Los Angeles, CA 90095, USA; 6Department of Hematology, Shenzhen People’s Hospital, Shenzhen 518020, China; 7Department of Hematology, PLA General Hospital of Southern Theater Command, Guangzhou 510010, China

**Keywords:** microfluidic chip, chronic myelogenous leukemia, liquid biopsy, circulating biomarkers, precision medicine

## Abstract

**Simple Summary:**

Chronic Myelogenous Leukemia (CML) is a common blood cancer that involves the uncontrolled growth of certain blood cells. Treating this disease effectively requires doctors to detect and eliminate even the smallest traces of cancer cells, a process known as managing minimal residual disease. However, current methods to find these cells, like bone marrow tests, are not only uncomfortable for patients but also have some drawbacks, such as missing cancer cells. This study introduces an innovative, less invasive approach using a microfluidic chip. This chip works by analyzing a patient’s blood sample to find and measure the number of leukemia cells. We tested this device on 56 patients with CML and found that it could successfully identify these cancer cells in blood, with a high accuracy rate. This new method is promising because it is less invasive and more precise, making it easier to monitor how well a patient’s treatment is working and to make better treatment decisions. This advancement could significantly improve the care and outcomes of people with CML, making a real difference in their lives.

**Abstract:**

Chronic Myelogenous Leukemia (CML) is a prevalent hematologic malignancy characterized by the malignant transformation of myeloid cells and their proliferation in the peripheral blood. The management of CML poses significant challenges, particularly in detecting and eradicating minimal residual disease, which is crucial for preventing relapse and improving survival outcomes. Traditional minimal residual disease detection methods, such as bone marrow aspiration, are invasive and have limitations which include the potential for sampling errors and false negatives. This study introduces a novel label-free microfluidic chip designed for the segregation and recovery of circulating leukemia cells, offering a non-invasive liquid biopsy approach with potential applications in precision medicine. Over July 2021 to October 2023, we recruited 56 CML patients across various disease stages and collected blood samples for analysis using our microfluidic device. The device demonstrated high efficacy in isolating circulating leukemia cells, with an optimal capture efficiency of 78% at a sample flow rate of 3 mL/h. Our results indicate that the microfluidic device can efficiently segregate and quantify circulating leukemia cells, providing a detailed understanding of CML progression and treatment response. The significant reduction in circulating leukemia cell counts in patients in complete remission highlights the device’s potential in monitoring treatment efficacy. Furthermore, the device’s sensitivity in detecting minimal residual disease could offer a more reliable prognostic tool for therapeutic decision-making in CML management.

## 1. Introduction

Chronic Myelogenous Leukemia (CML) is a common hematologic cancer which involves malignant transformation of myeloid cells inside the bone marrow that then spread out to the peripheral blood. The incidence of CML has been on an upward trajectory, particularly in correlation with the aging demographic of the population. Data from the National Cancer Institute elucidate that CML continues to pose significant clinical challenges, with an estimated 8930 new cases and approximately 1310 fatalities in the United States for the year 2023 [1]. Despite the emergence of targeted therapy and the implementation of allogeneic hematopoietic stem cell transplantation, the disease is plagued by high rates of relapse, especially pronounced in the older patient demographic [2,3,4]. A key challenge in the management of CML is the timely detection and subsequent eradication of minimal residual disease (MRD), defined as the residual leukemic cells that persist post-treatment. MRD serves as a robust prognosticator for relapse and is directly correlated with overall survival outcomes [5,6].

Traditionally, MRD in CML patients has been assessed using invasive procedures such as bone marrow aspiration, followed by cytogenetic or molecular techniques. Fluorescent in-situ hybridization (FISH) and Polymerase Chain Reaction (PCR) are commonly employed to detect BCR-ABL fusion genes or transcripts, which are the distinctive biomarkers of CML [7,8]. While these methods are highly sensitive and specific, bone marrow biopsies are invasive, can cause discomfort to the patient, and are challenging to perform frequently. This is a significant drawback, especially when continuous monitoring is required to assess disease progression or response to therapy. The heterogeneous distribution of leukemia cells within the bone marrow further complicates MRD detection, leading to potential sampling errors. A sample may not be representative of the overall burden of disease, resulting in inaccurate assessments and potentially misguided clinical decisions [9]. In light of these challenges, there is an urgent need for more convenient, non-invasive, and sensitive approaches to monitor MRD in CML. Peripheral blood-based MRD assessment is emerging as a promising alternative, offering a less invasive method for frequent monitoring.

Recently, the detection of circulating leukemia cells (CLCs) in peripheral blood has emerged as a promising strategy for the assessment of MRD in CML. Studies have consistently demonstrated a strong correlation between the presence of CLCs and both disease progression and MRD burden, highlighting their potential as biomarkers [10,11]. In contrast to bone marrow samples, peripheral blood can be collected through non-invasive means and subjected to frequent analysis, thereby facilitating continuous monitoring of the patient’s disease status. However, the extremely low concentration of CLCs in the bloodstream, ranging from 1 to 100 cells per 106 normal blood cells depending on the stage of the disease, poses a significant challenge for detection [12]. Multiparameter flow cytometry (MFC) has demonstrated potential in identifying these rare cell populations within blood samples. However, some inherent limitations restrict its efficiency and broader application. One of the primary issues is the background noise generated during analysis, which can obscure the detection of rare cell events, leading to potential inaccuracies [13]. Additionally, the red blood cell lysis phase, a crucial step in MFC, can inadvertently result in the loss of the targeted CLCs, further affecting the sensitivity of MFC. These challenges underscore the need for a more refined and sensitive detection method.

Advancements in microfluidic technologies have opened a new era for the isolation and analysis of rare circulating cells, demonstrating substantial promise and capabilities [14,15]. Previous research has demonstrated the capability of microfluidic chips to provide valuable insights into treatment responses across a variety of cancers [14,15,16]. Within the realm of microfluidic technologies, affinity-based sorting has established itself as a fundamental cell sorting technique [17,18,19]. Jackson et al. pioneered using an antibody-affinity-based microfluidic chip for isolating circulating leukemia cells in Acute Myeloid Leukemia (AML). This device, utilizing a CD34 antibody, selectively isolates circulating leukemia cells from the peripheral blood of AML patients [20]. This innovative method has yielded between 60 to 200 CLCs per mL of blood with a purity range of 2–6% from the patients with active disease stage. Despite its promising outcomes, the CD34 antibody-based microfluidic device is confronted with several challenges. The mechanism of affinity-based sorting can lead to non-reversible binding, posing difficulties in cell retrieval without compromising cell viability [21,22,23]. Moreover, this method relies on the specific binding of antibodies to antigens on the surface of target cells. If the antigen is also present on non-target cells or if the antibody has cross-reactivity with other antigens, it can lead to false positives and contamination of the sorted cell population [24]. Furthermore, the complex fluid dynamics inherent in the process can influence the efficiency of cell capture, with the risk of reduced interaction times at high flow rates and the potential for clogging at low flow rates [25].

To address the limitations associated with affinity-based sorting methods, our study used the specially patterned micropillars microfluidic assay, a technique previously validated in other rare cell sorting applications [26,27]. In this work, we developed a label-free microfluidic device, meticulously tailored for the isolation of circulating leukemia cells from blood, capitalizing on their distinct physical and biomechanical characteristics. A noteworthy achievement of our device is its capability to recover isolated circulating leukemia cells, facilitating subsequent cytogenetic analyses. This capability provides clinicians with a profound insight into the characteristics of circulating leukemia cells, thereby aiding in the formulation of personalized treatment strategies. Utilizing this novel device, we have conducted a comprehensive evaluation of MRD status across various stages of CML, comparing our results with those obtained from current clinical prognostic tools, including bone marrow evaluations conducted via multiparameter flow cytometry (MFC). The results showed that the MRD tracking facilitated by our microfluidic assay correlates well with both the administered therapeutic treatments and the observed patient outcomes. In the longitudinal study, we even noticed the microfluidic device can catch the elevation of CLC levels much earlier than the MFC, heralding an early indication of disease relapse. This underscores the potential of our microfluidic device as a more sensitive tool for disease monitoring and management.

## 2. Material and Methods

### 2.1. Cultivation of Leukemia Cancer Cell Line 

In our study, we utilized the K562 CML cell line (American Type Culture Collection). The cells were maintained in suspension culture using RPMI-1640 medium (Gibco™, Newburyport, MA, USA), supplemented with 10% fetal bovine serum (FBS, Gibco™, Newburyport, MA, USA). Optimal growth conditions were ensured by incubating the cells at 37 °C in a controlled atmosphere comprising 5% carbon dioxide. To maintain cellular vitality and ensure consistent experimental results, the cells were sub-cultured every 3 days, with careful monitoring to keep them within the 8th to 14th generations. 

### 2.2. Clinical Sample Collection

The research protocol was reviewed and approved by the Ethics Committee of the General Hospital of Southern Theater Command, and written consent was obtained from all participants, including patients and healthy donors. Over the span of July 2021 to October 2023, we successfully recruited a diverse cohort of 56 CML patients, representing various disease stages (newly diagnosed, remission, relapsed). Blood samples were collected from each participant and preserved in Vacutainer tubes (BD-Becton Dickinson, Franklin Lakes, WA, USA), which contained a stabilizing solution of ethylenediaminetetraacetic acid (EDTA). To ensure the preservation of sample integrity and the accuracy of subsequent analyses, all freshly obtained blood samples were maintained at ambient room temperature on a nutator. This practice facilitated gentle mixing and prevented coagulation. Importantly, to maximize the reliability of our experimental results, all laboratory experiments were conducted within one hour of blood sample collection.

### 2.3. Microfluidic Device Design Principle to Separate CLCs from Blood

The microfluidic device illustrated in Figure 1 is ingeniously designed to segregate CLCs from blood, and it operates through two main stages. The first stage of the device incorporates an array of pillars arranged in a deterministic lateral displacement (DLD) pattern; a design principle that has been refined based on our previous research endeavors [28]. This DLD array is crucial for the selective removal of smaller blood components, such as red blood cells (RBCs) and platelets, ensuring that only larger cells like normal white blood cells (WBCs) and leukemia cells are allowed to proceed to the second stage of the device.

Leukemia cells are known to exhibit unique biomechanical properties that distinguish them from regular WBCs. A study conducted by Zhou et al. employed optical tweezers to meticulously measure the elastic modulus of various human leukemia cell types and neutrophils, including macrophages, monocytes, and granulocytes [29]. Their results demonstrated that chronic myelogenous leukemia (K562) cells have an elastic modulus that is 2.5 to 4 times higher than that of macrophages, monocytes, and granulocytes, with measured values of E = 75 ± 33 Pa, E = 33 ± 13 Pa, E = 29 ± 13 Pa, and E = 25 ± 11 Pa, respectively. Leveraging these distinctive biomechanical characteristics, our microfluidic device is specifically engineered to separate leukemia cells from normal WBCs in the second stage. The device features a complex arrangement of micropillars, with every three pillars forming a single trapping unit that acts as a selective filter. By precisely tuning the spacing between these micropillars and carefully controlling the flow rate of the fluid through the device, we create conditions that allow the more flexible WBCs to navigate through the gaps between the pillars, while the stiffer leukemia cells are effectively trapped. Our previous work resulted in a computational model that identifies an optimal micropillar spacing of 11 μm and a flow rate of 6 mm/s for this purpose [27]. This specific configuration ensures that the majority of WBCs are filtered out, leaving the leukemia cells isolated within the trapping units, ready for subsequent analysis and study.

### 2.4. Microfluidic Assay for CLCs Separation, Enumeration, and Retrieval

Our microfluidic device is designed to utilize the unique size and stiffness characteristics of CLCs. We first accessed the device’s efficacy in isolating CLCs at various flow rates. To do this, we mixed approximately 500 K562 cells with 1 mL of peripheral blood from healthy donors to mimic the scenario of a CML patient’s sample, and then ran through the microfluidic channel to test the devices’ ability to isolate circulating leukemia cells. The sample was introduced into its designated inlet, while the buffer was simultaneously infused into the side inlet (Figure 1A). Both components were administered at varied flow rates to ensure their convergence at the central point of the chip. A blood sample without pipetting K562 cells was also tested as a negative control. Finally, blood samples from actual CML patients at various disease phases—newly diagnosed, relapsing, and in remission—were employed to validate the device’s capability in tracking disease progression and assessing the efficacy of treatment responses.

Post-capture, the cells underwent fixation with 5% paraformaldehyde and were permeabilized using a 0.1% Triton X-100 (Sigma-Aldrich, St. Louis, MO, USA) solution in PBS. This was followed by a blocking phase using 5% FBS (Wisent, Saint-Jean-Baptiste, QC, Canada) for 20 min. The CLCs were subsequently identified through a meticulous staining protocol, employing 40,6-diamidino-2-phenylindole (DAPI, eBioscience, San Diego, CA, USA), PE-Cyanine3 conjugated with CD33, CD13 human monoclonal antibody (eBioscience, San Diego, CA, USA), Alexa Fluor 647-associated CD45 human monoclonal antigen (eBioscience, San Diego, CA, USA), and the negative aberrant markers FITC-conjugated CD117 and CD34 human monoclonal antibodies (manifesting as DAPI+, CD33 or CD13, CD45+, CD117− and CD34−) [30,31]. The images in Figure 2 were captured utilizing the Zeiss Axio Observer inverted fluorescence microscope, followed by the in-chip enumeration of CLCs to assess tumor burden. 

For the retrieval assay, post-capture, a reverse flow was initiated from the lateral flow channel using a syringe pre-filled with PBS buffer. The entrapped cells were then flushed out at a flow rate of 5 mL/h for a duration of 1–2 min (Figure 1B), rendering them ready for subsequent cellular and molecular analyses.

### 2.5. Bone Marrow Aspiration Sample Accessed by Multi-Parameter Flow Cytometry (MFC)

Bone marrow aspirates accessed by multi-parameter flow cytometry (MFC) is a powerful technique in clinical practice for detecting MRD and quantifying tumor burden in CML patients [32,33]. This method not only facilitates the detailed analysis of cellular components but also enables the physical segregation and collection of specific cell populations. It provides a critical benchmark, allowing us to evaluate the diagnostic performance of our innovative microfluidic assays in comparison to this well-established standard.

Bone marrow aspirate samples are processed using a six-color clinical flow cytometry system (BD MFCCanto™). Mononuclear cells are extracted from a 5 mL bone marrow aspirate sample utilizing the Ficoll gradient centrifugation technique. Following this, an erythrocyte lysis step is conducted to eliminate any remaining red blood cells. To precisely identify leukemia cells within this diverse cell mixture, the harvested cells are incubated with a carefully selected panel of fluorescent dye-conjugated antibodies, targeting CD13, CD33, CD34, CD45, and CD117.

After the incubation period, the labeled cells are loaded into the MFC. The BD MFC Canto^TM^ Clinical Software, V3.0 is then employed to process the signals, effectively distinguishing the leukemia cells based on their unique immunophenotypic characteristics, defined as CD33+ or CD13+, CD45+, CD117− and CD34− [30]. Following their identification through fluorescence and light scatter properties, the leukemia cells are sorted into separate containers, making them available for downstream analyses.

### 2.6. Fluorescence In Situ Hybridization (FISH) Assay

CML is characterized by the presence of the Philadelphia chromosome (Ph), a cytogenetic abnormality resulting from the reciprocal translocation t(9;22)(q34;q11). This translocation fuses the BCR gene on chromosome 22 to the ABL1 gene on chromosome 9, creating the BCR-ABL1 oncogene [34]. Fluorescence in situ hybridization (FISH) is a robust and sensitive cytogenetic technique employed to detect this specific genetic abnormality in CML. Following the recovery of circulating leukemia cells (CLCs), a gradient centrifugation step is performed to purify the cell suspension and discard the supernatant. The cells are then fixed with 100% ethanol for a duration of 5 min and meticulously transferred onto a glass slide. The slide is then heated at 75 °C for 30 min. Following this, the slides are subjected to a denaturation process in a 70% formamide/2× Tween solution at 72 °C for 3 min and then dehydrated through a series of ethanol washes (70%, 85%, and 100%) for 2 min each, until completely dried. For the detection of the BCR-ABL fusion gene, BCR-ABL1 fusion FISH probes (Fisher Scientific, Hampton, MA, USA) were used to test any chromosome abnormality in the sample. An amount of 4 μL of probe per sample was mixed with 8 μL of hybridization diluent and denatured in a water bath at 75 °C for 6 min. The hybridization step involved applying the diluted probe reagent to the cell region on the glass slide, which was then covered and sealed with rubber cement to prevent evaporation. The samples were placed in a humidified environment at 45 °C to allow for overnight hybridization, facilitating the binding of the probes to the target DNA sequences. Post-hybridization, the coverslip was gently removed, and the slides underwent a series of washes in 0.4%× Tween/0.3% Tween at 66 °C for 2 min, followed by a wash in 2%× Tween/0.1% Tween at room temperature for 1 min. The slides were then left to air-dry. DAPI staining was applied to the hybridization area to counterstain the DNA, enhancing the visibility of the chromosomes under a fluorescence microscope. The final step involved loading the slides into the fluorescence microscope for imaging analysis, where the presence of the BCR-ABL fusion gene could be confirmed, providing crucial information for the diagnosis, prognosis, and monitoring of treatment response in patients with CML. This precise detection of cytogenetic abnormalities through the FISH assay is instrumental in the comprehensive management of CML, ensuring timely and targeted therapeutic interventions.

## 3. Results

### 3.1. Microfluidics Device Efficacy in Isolating CLCs across Different Flow Rates

We conducted experiments to evaluate the isolation efficiency of CLCs using a microfluidics device, varying the sample flow rates from 1 to 9 mL/h in 2 mL/h increments, while concurrently adjusting the buffer flow rates from 1.2 to 9.2 mL/h. To ascertain the optimal flow rates for CLC isolation, we employed a fluorescence microscope to meticulously count the K562 cells. The formula used to calculate the isolation efficiency was:Number of K562 cells captured in the microfluidic chipTotal K562 cells injected into the sample×100% 

We conducted three trials for each set of flow rate parameters, yielding the following isolation efficiency results: 85.2 ± 5.3%, 78.5 ± 4.2%, 50.8 ± 3.2%, 28.4 ± 6.3%, and 8.5 ± 4.0%. A noticeable trend emerged, indicating a dramatic decrease in CLC capture efficiency as the flow rate increased. To set up an optimal balance between time efficiency and capture rate, we selected a sample flow rate of 3 mL/h and a buffer rate of 3.2 mL/min, achieving an approximate capture efficiency of 78%, as illustrated in Figure 3. This configuration ensures a high rate of CLC isolation while maintaining a reasonable processing time to avoid clogging, highlighting the microfluidics device’s capability in efficient cell separation under varied flow conditions.

### 3.2. Chronic Myelogenous Leukemia Patient Samples Processing in Microfluidic Device through Different Disease Stages

First, we aimed to validate the clinical applicability and precision of our innovative label-free microfluidic chip in segregating CLCs from blood samples of patients at varying stages of CML. A total of 56 blood specimens were meticulously analyzed, representing a broad spectrum of disease stages. This cohort included 26 samples from patients in the active disease status of CML, either at newly diagnosis or during a relapse, categorized as the high tumor burden group. Additionally, 14 samples were obtained from patients who had achieved complete remission, representing a low tumor burden scenario. Another 16 samples were collected from patients undergoing treatment but in a state of partial remission. To establish a baseline and for comparative purposes, four blood samples from healthy donors were also processed as negative controls.

Following the cell capture within the microfluidic chip, we utilized an immunofluorescence assay for the identification and quantification of the isolated CLCs. The results, illustrated in Figure 4A, showed that patients in the newly diagnosed or relapsing stage exhibited an average CLC count of 99/mL, with a range spanning from 15 to 344/mL. In contrast, patients who had achieved complete remission displayed significantly lower CLC counts, ranging from 0 to 4/mL. Those in partial remission presented with CLC counts varying between 1 and 21/mL. The healthy donors, as anticipated, demonstrated the lowest CLC counts, fluctuating between 0 and 2 CLCs/mL.

Through statistical analyses, we confirmed the device’s efficacy in differentiating between active disease stages and remission stages, yielding a highly significant *p*-value of *p* < 0.0001. Additionally, the CLC counts in symptomatic CML patients under treatment were significantly distinct from those in healthy controls, with a *p*-value of *p* < 0.0001. The Receiver operating characteristic (ROC) analysis in Figure 4B established a threshold of six CLCs to differentiate active disease from remission with a 99% confidence interval, achieving a sensitivity of 100% and a specificity of 93%. The chance of not detecting active disease when it is present (false negative rate) was only 7%, and we did not find any cases where the disease was wrongly identified in healthy individuals (no false positives). The average purity of isolated CLCs from symptomatic CML patients, estimated at 58%, indicates a satisfying level of specificity in isolating the target cell population, though there is room for improvement to enhance the purity and overall efficiency of the device.

### 3.3. Cytogenetic Analysis of Circulating Leukemia Cells by FISH Assay

The application of our novel microfluidic device for the separation of CLCs from peripheral blood represents a significant advancement in the cytogenetic analysis of CML. The step of separation essentially enriches the concentration of CLCs, which reduced background noise and improved the signal-to-noise ratio in subsequent FISH assays. Figure 5A demonstrated here revealed the presence of the BCR-ABL1 fusion signal as juxtaposed red and green signals, indicating a fusion of the BCR and ABL1 gene regions. Conversely, Figure 5B illustrates a normal cell where two separate red and green signals were observed, representing the normal status of BCR and ABL1 genes. This methodological enhancement led to an increase in the detection sensitivity of the BCR-ABL1 fusion signal, with a higher percentage of cells positive for the fusion gene observed compared to conventional FISH assays performed on unsorted samples. The microfluidic device facilitated the identification of the BCR-ABL1 fusion in a greater proportion of patient samples, including those with low disease burden, thereby demonstrating the potential for earlier and more accurate diagnosis.

### 3.4. Longitudinal Monitoring of Leukemic Cell Levels in CML Using Microfluidic Platform and Multiparameter Flow Cytometry

To ascertain the effectiveness of our microfluidic chip in monitoring disease progression and tumor burden in CML patients, we conducted a longitudinal study with individuals undergoing follow-up treatment. We measured CLCs at different stages of the disease and compared these measurements with bone marrow aspirate results analyzed through MFC. Representative MFC test images are provided in Figure 6 for reference.

As illustrated in Figure 7A, Patient 1, who was initially treated with Imatinib, a first-line tyrosine kinase inhibitor (TKI) specific for the BCR-ABL1 fusion protein in CML, showed a decrease in CLC counts from 105 to 7 per mL over a 45-day period, as determined by our microfluidic device followed by FISH testing. Concurrently, MFC tests indicated a reduction in leukemic cell counts from 327 per 106 nucleated cells to 0, suggesting a favorable response to TKI therapy. However, after a period of 60 days, an increase in CLC counts was observed, suggesting potential resistance to Imatinib. Such elevation was not detected in bone marrow aspirate until two months later. The patient was then switched to a second-generation TKI, Dasatinib, which led to undetectable CLC levels and leukemic cell counts in bone marrow post 45 days of treatment, indicating a status of remission.

Patient 2 was administered Nilotinib after initial diagnosis, another second-generation TKI. Within the first 30 days, CLC counts decreased significantly from 142 to 13 per mL of blood, reflecting the decrease in tumor burden also observed in bone marrow aspirates analyzed by MFC (Figure 7B). Over the next 60 and 180 days, CLC counts remained undetectable, consistent with a state of complete remission.

## 4. Discussion

The results from our study highlight the microfluidic device’s capability in efficiently segregating and quantifying CLCs across different stages of CML, providing a detailed understanding of the disease’s progression and response to treatment. The significant reduction in CLC counts in patients who have achieved complete remission compared to those in active disease stages or partial remission underscores the device’s potential in monitoring treatment efficacy and disease progression. 

Detecting MRD, as evidenced by the low CLC counts in patients in complete remission, is crucial in CML management, aiming often to achieve and maintain a deep molecular response. The statistically significant difference in CLC counts between the remission group and healthy donors aligns with clinical definitions of remission, where the blood is expected to be devoid of leukemia cells [35]. This observation adds a layer of validation to our microfluidic device, demonstrating its potential as a reliable tool in the clinical setting for monitoring CML patients across different disease stages, aiding in timely and accurate therapeutic decision making. However, it also raises intriguing questions about the nature of complete remission in CML, questioning whether it truly represents a disease-free state or if there are residual leukemia cells that remain undetectable by current methodologies. Future endeavors could focus on further optimizing the device’s efficiency and exploring its applicability in other hematological malignancies.

The integration of a microfluidic device into the FISH assay workflow for CML represents a paradigm shift in cytogenetic analysis. Traditional FISH assays, while sensitive, are limited by the prevalence of non-leukemic cells in peripheral blood samples, which can obscure the BCR-ABL1 fusion signal [36]. Our microfluidic approach addresses this limitation by pre-selecting a population of cells with a high likelihood of containing targeted mutated genes, thereby enhancing the sensitivity of the FISH assay. This methodological improvement is particularly impactful in the MRD detection, where the detection of low levels of leukemic cells is critical for patient management and prognosis. Furthermore, the specificity of cell capture by the microfluidic device reduces the likelihood of false positives, ensuring that the FISH signals observed are truly indicative of the presence of the BCR-ABL1 fusion gene. The ability to detect MRD with greater sensitivity and specificity allows for more accurate monitoring of treatment response and timely intervention upon disease progression or relapse.

Our longitudinal study underscores the microfluidic assay’s potential as a powerful, non-invasive tool for monitoring disease progression in CML patients. By evaluating CLC levels across different disease stages in patients and correlating these findings with bone marrow aspirates analyzed through MFC, we have demonstrated the assay’s effectiveness. In all cases, the CLC counts detected via the microfluidic platform showed a high correlation with results from bone marrow aspirates, confirming the platform’s reliability. Notably, the microfluidic assay’s capacity to detect early changes in CLC levels, potentially even before changes are observable in bone marrow, underscores its critical role in proactive disease management. Furthermore, it overcomes the inherent limitations associated with bone marrow aspirates, such as sampling errors and the heterogeneous distribution of leukemia cells within the bone marrow, which can lead to false negatives during early relapse stages [37].

## 5. Conclusions

In conclusion, our microfluidic chip offers a non-invasive, sensitive, and precise tool for the management of CML. It holds the potential to revolutionize the monitoring of disease progression and response to treatment, ensuring timely and targeted therapeutic interventions. Future studies and clinical trials will be conducted in further validating the clinical applicability of this technology and its integration into standard CML management protocols.

## Figures and Tables

**Figure 1 cancers-15-05696-f001:** Schematic of the CLCs microfluidic device. (**A**) The device is composed of a two-stage cell separation system. The first stage employs a Deterministic Lateral Displacement (DLD) array, consisting of a series of micropost arrays. Particles with smaller diameters, such as red blood cells (RBCs) and platelets, would be separated from nucleated cells. The second stage utilizes a custom arrangement of micropillars that are engineered to capture leukemia cells by its distinct elastic modulus, while allowing the more flexible normal white blood cells to pass. (**B**) The CLCs are retrieved by applying a reverse flow through a lateral channel, allowing for their collection at a designated side outlet for further analysis.

**Figure 2 cancers-15-05696-f002:** Representative images showing circulating leukemia cells trapped by micropillars and immunostaining for identification using appropriate filter cubes. (**A**) DAPI+, stained in blue; (**B**) CD33+ or CD13+ by PE-Cyanine3, stained in yellow; (**C**) both negative for aberrant makers CD34 and CD117 by FITC, stained in green and (**D**) CD45+ by Cy5, stained in red.

**Figure 3 cancers-15-05696-f003:** Evaluation of the microfluidic device’s performance in isolating leukemia cells across different sample flow rates, specifically at 1 mL/h, 3 mL/h, 5 mL/h, 7 mL/h, and 9 mL/h. Each flow rates repeats 3 times. Approximately 500 K562 cells were introduced into 1 mL of blood from a healthy donor for assessment. The capture efficiency was calculated based on the proportion of K562 cells retained within the microfluidic channel in relation to the overall number of K562 cells initially incorporated into the blood specimen. Each flow condition was repeated three times.

**Figure 4 cancers-15-05696-f004:** (**A**) The enumeration of CLCs isolated through our label-free microfluidic chip from patient peripheral blood samples. Each dot represents the CLCs count of each patient. In newly diagnosed or relapsed CML patients, CLCs ranged from 15 to 344 per mL, showing a markedly higher count than that in patients in remission (*p* < 0.0001). A clear statistical distinction was also observed between patients in complete remission and those in partial remission (*p* < 0.0001). On the other hand, both complete remission patients and healthy donors exhibited extremely low levels of CLCs, ranging from 0 to 4/mL, and no significant statistical difference was found between these two groups. (**B**) Receiver operating characteristic (ROC) analysis of CLC counts between CML patients in active disease stage and remission stage. Each point on the ROC curve signifies a progressively higher threshold value. A threshold of 6 CLCs/mL was set to achieve a sensitivity of 100% and a specificity of 93%.

**Figure 5 cancers-15-05696-f005:** FISH test with dual color and dual transfusion probes for BCR (green) and ABL1 (red) to detect reciprocal translocation t(9;22)(q34;q11). (**A**) CLCs recovered from Patient 1 who had shown resistance to tyrosine kinase inhibitor therapy at 105 days of treatment. BCR-ABL translocation was detected in 24% of the 500 nuclei counted. (**B**) In contrast, a normal cell shows two separate sets of red and green signals (2R2G).

**Figure 6 cancers-15-05696-f006:** MFC to access minimal residual disease status from Patient 1’s bone marrow aspirate post 60 days of tyrosine kinase inhibitor therapy. The circles represents the gating process of Chronic myeloid leukemia cells, which are identified as (**A**) gating on nucleated cells using CD45(Red)/SSC(Green), a total of 107  nucleated cells were examined, fixed gating for expressing at CD33(Green) but negative for (**B**) CD117(Blue), and (**C**) CD34(Red); and fixed gating for expressing at CD13(Red) but negative for (**D**) CD117 and (**E**) CD34.

**Figure 7 cancers-15-05696-f007:** A longitudinal follow-up study was conducted to assess leukemia disease levels using both bone marrow aspirates processed by MFC and peripheral blood samples processed by a microfluidic device. (**A**) Patient 1, after undergoing TKI therapy, exhibited a significant reduction in both CLC levels and leukemia cell count in the bone marrow 45 days post-treatment, but later revealed Imatinib resistance with rising CLC counts undetected in bone marrow for two months. Switching to Dasatinib led to remission within 45 days. (**B**) Patient 2’s CLC count fell from 142 to 13 per mL within 30 days of starting Nilotinib, mirroring a similar decline in tumor burden as confirmed by MFC in bone marrow samples. Subsequent assessments at 60 and 180 days showed sustained undetectable CLC levels, indicative of sustained complete remission.

## Data Availability

The data presented in this study are available on request from the corresponding author. The data are not publicly available due to privacy restrictions.

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
