# Peer review of "Precision Isolation of Circulating Leukemia Cells in Chronic Myelogenous Leukemia Patients Using a Novel Microfluidic Device and Its Clinical Applications"

_cancers, 2023, doi:10.3390/cancers15235696_

Round 1
Reviewer 1 Report
Comments and Suggestions for Authors
This study profiled the efficacy of a novel label-free microfluidic chip in the isolation and quantification of CLCs from patients with Chronic Myelogenous Leukemia. There are several suggestions:As a new screening and diagnostic method, I suggest that it is more intuitive to use statistical methods to generate data to show the efficacy of microfluidic devices. For example, the specificity, accuracy, positive and negative rate. Statistical differences should be compared with other screening methods.
Comments on the Quality of English LanguageMinor editing of English language required
Author Response
Thank you for your valuable feedback. In response to your suggestion, we have incorporated additional statistical data in section 3.2 and marked in red to demonstrate the efficacy of our microfluidic device more clearly.
Reviewer 2 Report
Comments and Suggestions for Authors
Comentarios sobre el manuscrito ID cancers-2732384 "Precision Isolation of Circulating Leukemia Cells in Chronic Myelogenous Leukemia Patients Using a Novel Microfluidic Device and its Clinical Applications"
This article describes the performance and validation of a new device that improves the sensitivity and specificity of conventional methods to test for the presence of circulating leukemic cells by a first isolation step. The evidence presented is eloquent and the methodology is described in detail. I consider that their implementation will be of great use to specialists with positive consequences for patients with this type of neoplasm. My only observation is with respect to the references since they were not added in a proper format, so they can be accepted after minor revision.
Author Response
Thank you for the comment. We have adjusted our references format accordingly.
Reviewer 3 Report
Comments and Suggestions for Authors
Ouyang and colleagues have introduced a label-free microfluidic chip for non-invasive management of Chronic Myelogenous Leukemia (CML). Recruiting 56 CML patients, the chip demonstrated a 78% capture efficiency, effectively isolating and quantifying circulating leukemia cells. Its potential in monitoring treatment efficacy and serving as a reliable prognostic tool underscores its significance in advancing CML research. Minor points:
1. In the abstract (line 9) and the material and methods (item 2.2): change “chronic leukemia” to “Chronic Myelogenous Leukemia”. The term “chronic leukemia” alone can include chronic lymphocytic leukemia and confuse readers.
2. Use HUGO nomenclature for genes (Bruford et al, Leukemia 35:3040–3043, 2021). For example use “BCR::ABL1” instead of “BCR-ABL”.
3. In item 2.1., replace “K562 chronic leukemia cell line” with “K562 chronic Myelogenous leukemia cell line”
4. In the Conflicts of Interest item. Do the authors intend to or have submitted the method for patent? Is there any intention to commercialize the device in the future? If requested, will the authors make details about the manufacture of the device freely available to other academics? These aspects need to be clear in the "Conflicts of Interest" section to avoid a potential retraction of the article in the future.
Author Response
We really appreciate your support to our work and thank you for your valuable feedback. Please see our responses below.
- In the abstract (line 9) and the material and methods (item 2.2): change “chronic leukemia” to “Chronic Myelogenous Leukemia”. The term “chronic leukemia” alone can include chronic lymphocytic leukemia and confuse readers.
Thank you for your keen observations. We have modified accordingly.
- Use HUGO nomenclature for genes (Bruford et al, Leukemia 35:3040–3043, 2021). For example use “BCR::ABL1” instead of “BCR-ABL”.
Thank you for your suggestion. We have change to the HUGO nomenclature as requested.
- In item 2.1., replace “K562 chronic leukemia cell line” with “K562 chronic Myelogenous leukemia cell line”
Thank you for pointing it out, we have corrected it.
- In the Conflicts of Interest item. Do the authors intend to or have submitted the method for patent? Is there any intention to commercialize the device in the future? If requested, will the authors make details about the manufacture of the device freely available to other academics? These aspects need to be clear in the "Conflicts of Interest" section to avoid a potential retraction of the article in the future.
Thank you for the suggestion. All the authors agreed and there is no conflict of interest to declare.